# A Graph-Based Superpixel Segmentation Approach Applied to Pansharpening

**DOI:** 10.3390/s25164992

**Published:** 2025-08-12

**Authors:** Hind Hallabia

**Affiliations:** 1UMR CNRS 7347—Materiaux, Microéléctronique, Acoustique, Nanotechnologies (GREMAN), Institut Universitaire de Technologie de Blois (IUT Blois), Tours University, 37000 Tours, France; hind.hallabia@univ-tours.fr; 2Institut National des Sciences Appliquées Centre-Val de Loire (INSA CVL Campus Blois), 41000 Blois, France

**Keywords:** simple linear iterative clustering (SLIC), region adjacency graph (RAG) merging, graph-based superpixels, regression analysis, simplex algorithm

## Abstract

In this paper, an image-driven regional pansharpening technique based on simplex optimization analysis with a graph-based superpixel segmentation strategy is proposed. This fusion approach optimally combines spatial information derived from a high-resolution panchromatic (PAN) image and spectral information captured from a low-resolution multispectral (MS) image to generate a unique comprehensive high-resolution MS image. As the performance of such a fusion method relies on the choice of the fusion strategy, and in particular, on the way the algorithm is used for estimating gain coefficients, our proposal is dedicated to computing the injection gains over a graph-driven segmentation map. The graph-based segments are obtained by applying simple linear iterative clustering (SLIC) on the MS image followed by a region adjacency graph (RAG) merging stage. This graphical representation of the segmentation map is used as guidance for spatial information to be injected during fusion processing. The high-resolution MS image is achieved by inferring locally the details in accordance with the local simplex injection fusion rule. The quality improvements achievable by our proposal are evaluated and validated at reduced and at full scales using two high resolution datasets collected by GeoEye-1 and WorldView-3 sensors.

## 1. Introduction

A wide variety of Earth observation imaging sensors are available to deliver images about the covered scene (e.g., Pléiades, GeoEye-1, GaoFen-2, and WorldView-2/-3). Most of these satellites can produce two kinds of images: a panchromatic (PAN), image which is a greyscale image with high spatial resolution; and a multispectral (MS) image, which is a multi-band image, usually, with four or eight spectral bands and low spatial resolution. Hence, to generate a more comprehensive synthetic image for such a scene, remote sensing data fusion could be used to resolve this issue. Particularly, fusion of PAN and MS images, referred to as pansharpening [1,2,3], regarded as a sub-topic of the data fusion field, may be applied. Thus, a spatially enhanced MS image can be synthesized, which is at the same resolution as the PAN image, which ideally exhibits the spectral proprieties of the original MS image. The approach can be used in several applications, such as change and anomaly detection and land cover monitoring and mapping.

### 1.1. Background

Over the two last decades, numerous pansharpening algorithms have been proposed [4,5,6,7,8,9,10,11,12]. Based on the way details are extracted, they are classified into three categories: Multi-Resolution Analysis (MRA), Component Substitution (CS), and Model-Based Methods (MMs). The CS group operates in a transformed domain by replacing a component of the original MS image (intensity or first principal component) with the histogram-matched PAN image to generate a high-resolution MS image. The details are therefore presented as a residual image between the synthetic component and the histogram-matched PAN image. Instances include Brovey (BT) [13] and its optimized version with haze correction (BT-H) [14], generalized intensity-hue-saturation (GIHS) [15,16], principal component analysis (PCA) [15], the Gram–Schmidt adaptive (GSA) [17] method, partial replacement adaptive CS (PRACS) [18], band-dependant spatial detail (BDSD) [19], and the non-linear intensity-hue-saturation method (NL-IHS) [20]. Regarding the MRA group, it enables inference of details, which are extracted by means of spatial filtering the PAN image into the up-sampled MS bands. Some examples of this class are as follows: Indusion [21], the Generalized Laplacian Pyramid (GLP) tailored to the Modulation Transfer Function (MTF) with additive [22], multiplicative [7,23,24], and projective [25,26] injection models, the à-trous wavelet transform (ATWT) [27], the Filter Banks-based Pansharpening (FBP) [6] method, and the Half-Gradient Morphological Operator (MF-HG) [28]. Furthermore, with the intention of enhancing the quality of the fused products, the principles of MRA and CS are combined together and hybrid versions are developed [29,30,31], such that the MRA processing is performed for the intensity [29,30] or the first component [31]. A popular example is the Additive Wavelet Luminance Proportion (AWLP) [29], in which the wavelet transform is processed in the IHS domain.

The third group recasts the pansharpening task as an inverse problem, which can be solved with the application of regularization methods [32], convolutional neural networks (CNN) [9,33], and deep learning [34,35]. Considering the VO-based techniques categories, a variational optimization problem is defined and solved based on an acquisition or representation model and introducing regularization terms in some cases [32]. As examples, the model-based fusion using PCA and wavelets (PWMBF) [36] is proposed by modeling and solving the pansharpening via a maximum a posteriori problem. A variational approach based on filter estimation, known as FE-HPM [37], models the pansharpening as a deconvolution task by estimating a spatial degradation filter approximating the MTF of the MS sensors and relaying on the reduced- and full-scale spatial resolutions acquisitions. Pansharpening based on Total Variation (TV) [38] is proposed in which the fusion is modelled as a TV-regularized least squares problem, and solved according to the majoration-minimization optimization algorithm. The sparse representation of injected details (CS-D) [39] is an example of a pansharpening technique based on sparse representation theory. The spatial details to be injected into the up-sampled MS bands by a conventional MRA schema are generated through a dictionary of patches defined at reduced resolution taking advantage of the invariance property. Deep learning-based pansharpening methods [35] estimate the spectral and spatial properties of the high-resolution fused products using non-linear relationships estimated between the high-resolution PAN image and the low-resolution MS image. Consequently, advanced models, such as conventional neural networks (CNNs) [3,33,40], generative adversarial networks (GANs) [41] and auto-encoders (AEs) [42] have been applied in the field of pansharpening. Among CNN-based approaches, Masi et al [33,43] proposed a CNN network (model) introduced into the fusion process, and the method is referred to as PNN. The high-resolution MS image is synthesized using an ad hoc generation framework. It may be noted that CNN-based pansharpening methods are deeply related to the training procedure, a fundamental step for learning. The training stage for PNN is closely based on a resolution downgrading procedure, guided by Wald’s protocol [44]. For learning and updating the network parameters, a fused MS-PAN composite is generated and used as input to the CNN model. A more advanced variant of PNN is introduced in [40] by including a fine-tuning step in the training procedure.

### 1.2. Related Works

Most of the aforementioned pansharpening algorithms are processed on the original pixels (i.e., at pixel-level) [2]. In some cases, they neglect local contextual information about the targeted objects contained in the scenes which can not be sufficiently represented semantically, mainly for textured remote sensing imageries [24]. To reduce these problems, recent research studies focus on processing pansharpening at a region level. In [7,11,45], the authors argue that the fused results at feature level are better than those processed on pixels. In this context, several forms of local feature information are incorporated into the GLP framework, such as sliding windows [22], segments [45], clusters [11] or superpixels [7]. This technique has demonstrated its effectiveness in providing high quality fused outcomes at feature level by considering multiple injection gains. In [22], the authors defined them by computing local statistics over sliding windows, such that the model is referred to as Context-based Decision (CBD). Intuitively, the fixed structure of the latter causes overlap between the blocs. As a consequence, the spatial arrangement of the objects contained in the remote sensing data is ignored. To solve this issue, image clustering has been applied for pansharpening. For instance, in [11], the k-means clustering algorithm is adopted to generate several clusters of pixels showing the same semantic proprieties. Then, the injection coefficients are estimated for each cluster either as a solution of a multiple [10] or a robust regression problem [11]. Moreover, as in [45], the gain factors are computed using a statistical (i.e., projective injection model) through regions defined by performing the binary partition tree (BPT) segmentation method, which are incorporated in both the GLP and GSA techniques (referred as GLP-BPT and GSA-BPT, respectively). In addition, in another recent study [7], the pansharpening task is performed at superpixel level. In detail, the local information is defined over a set of superpixels extracted by means of a Simple Linear Iterative Clustering (SLIC) method [46] and the fusion process is performed using an optimized injection scheme that optimally calculates the injection gains through Shuffled Complex Evolution. Finally, as shown in [47], context-adaptive pansharpening methods can provide better fusion outcomes in terms of spectral fidelity than global techniques. This is justified by the fact that the local structural characteristics of regions could be better expressed by a set of pixels having the same proprieties as single pixels. In consequence, region-based fusion models are more effective in pansharpening by considering extracted features.

### 1.3. Motivation and Contributions

Existing pansharpening approaches are based on a unified fusion model. In some cases, spatial or spectral distortions could occur in fused products due to the diversity of sensor characteristics. Considering that remote sensing images contain heterogeneous pixels, and therefore diverse spectra, region-based fusion rules allows adaptive fusion of images to enhance the spatial and spectral resolutions in comparison with pixel-wise methods. Motivated by this idea, we introduced spatially variant injection gains estimated over multiple superpixels and its variant based on graph theory. In light of this issue, processing pansharpening on graph-based superpixels entities allows us to approximate very-high-resolution images. In this paper, a graph-driven pan-sharpening method is proposed, which is reformulated as a local regression problem with the purpose of preserving spectral fidelity and increasing the spatial quality of the MS image. The fusion process is performed on various regions defined by a graph-based SLIC technique applied on the MS image. The MS image is encoded as a graph to be used further as a guidance map for inference of spatial details into the expanded MS bands. In more detail, the proposed approach utilizes graph-based segmentation images by adopting a superpixel aggregation similarity concept instead of using explicit superpixel entities. The underpinning idea is to over-segment the MS image into a set of superpixels by means of SLIC [46], which represent initialization for the Region Adjacency Graph (RAG) [48]; then, a hierarchical merging step is performed. A graph-driven image is therefore generated with a number of regions lower than those obtained in the initial over-segmentation. Furthermore, a simplex-based fusion strategy is adopted in this study, where the injection gains are estimated over the proposed graphical regions. The pan-sharpening task is accomplished by regionally inferring details into their corresponding up-sampled MS regions defined by the graph-based segmentation map. The proposed approach does not inject redundant information into the MS image thanks to the graph-based detail injection procedure. The GLP technique is utilized for information analysis and further in fusion processing because it helps to preserve the edge information and consequently the details to be extracted.

In summary, our main contributions are three-fold:Modeling the guidance image (for detail injection) as a graph through a region adjacency graph (RAG) performed over an initial over-segmentation obtained by means of SLIC and followed by a hierarchical merging step to obtain a meaningful regional map.A graph-driven pansharpening method is proposed by considering the local simplex algorithm for gain estimation as well as the graph-based guidance image (map) for detail injection, with the aim of refining the fused outcomes at region level.Extensive experimental comparisons between pixel- and region-wise pansharpening methods as well as between superpixel versus graph-driven superpixel guided injection schemes are presented. The performance of the proposed method is highlighted in comparison with traditional, variational optimization, and deep learning methods.

The paper is structured as follows. Section 2 gives an overview about the basic principles to be used in our study. The proposed graph-based pansharpening algorithm is presented in Section 3, including the generation of the guidance injection map using RAG and SLIC, the formulation of the graph-based fusion scheme within the Simplex model, and its introduction into the pyramidal framework. Section 4 describes the experimental results assessing the performance and robustness of the proposed fusion method. Finally, the conclusions and future perspectives are presented in Section 5.

## 2. Preliminaries and Background

### 2.1. Pansharpening as a Unified Model for Fusion

Classical pansharpening methods can be recast using a unified protocol, which can be defined by the following two phases: (i) detail extraction; and (ii) its subsequent inference into the MS channels [3,4]. Thus, the fusion process relies on the transfer of detail components, ${\left\{D^{i}\right\}}_{i=1}^{N}$ (where *N* is the total number of spectral bands) into the expanded MS channels, $\tilde{MS}={\left\{{\tilde{MS}}^{i}\right\}}_{i=1}^{N}$, preponderated by injection gains, and ${\left\{G^{i}\right\}}_{i=1}^{N}$. This allows generating an enhanced MS image at fine-resolution, $\hat{MS}={\left\{{\hat{MS}}^{i}\right\}}_{i=1}^{N}$, which is at the same resolution as the original PAN image and has a high spectral-spatial quality [49].

From a mathematical point of view, the fusion algorithm can be expressed by the following general band-wise equation [1]:(1)$${\hat{MS}}^{i}={\tilde{MS}}^{i}+G^{i}\circ D^{i},i=\left\{1,\ldots ,N\right\},$$
where ∘ is the pixel-wise multiplication operator.

For detail extraction, taking into account the multi-resolution (MRA) and composition substitution methods, ${\left\{D^{i}\right\}}_{i=1}^{N}$ can be derived as the image residual between the histogram-matched PAN image, ${\left\{P^{i}\right\}}_{i=1}^{N}$, and the low-resolution PAN image, ${\left\{P_{L}^{i}\right\}}_{i=1}^{N}$ (for MRA-based methods), or the intensity component, *I* (for CS-based methods). Concerning the VO- and DL-based methods, the high-frequency details can be extracted through a mathematical model, denoted by Φ, as a function of PAN and MS data before/and during the fusion procedure. Generally, the details can be defined as follows for each band (i={1,..,N}):(2)$$D^{i}=\left\{\begin{array}{cc} \left(P^{i}-P_{L}^{i}\right) & \mathrm{for}\;\mathrm{MRA}\\ \left(P^{i}-I\right) & \mathrm{for}\;\mathrm{CS}\\ \Phi (P^{i},MS^{i}) & \mathrm{for}\;\mathrm{VO}\;\mathrm{and}\;\mathrm{DL} \end{array}\right.$$
where Φ(.,.) denotes the mathematical model or function related to PAN and MS data fusion.

Concerning the detail injection stage, ${\left\{D^{i}\right\}}_{i=1}^{N}$ can be inferred into the expanded MS bands ${\left\{{\tilde{MS}}^{i}\right\}}_{i=1}^{N}$ modulated by means of gain coefficients ${\left\{G^{i}\right\}}_{i=1}^{N}$ [47]. The latter allows proportionally inferring the spatial details into the expanded MS bands taking into consideration several forms of injection strategies at different fusion levels [50] in a multiplicative or projective manner [1,7,11]. In this context, there are a large variety of pansharpening methods which use either a global or a context-adaptive injection model.

#### 2.1.1. Global Injection Scheme

Fusion techniques that employ a global additive injection scheme consider a unique gain coefficient per spectral MS band or for the whole image. Instances include the GIHS [51], GLP with MTF-adjusted filters (GLP-HPF) (where HPF stands for high-pass filtering), AWLP [29], BDSD [19], and PRACS [18]. The two latter methods fall within the band-level fusion category since a unique gain factor is applied for each spectral channel.

#### 2.1.2. Multiplicative Injection Scheme

This model is also known as High-Pass Modulation (HPM), which is generally performed at pixel level [1]. The gain factors are computed for each pixel in the image as the ratio between the expanded MS image ($\tilde{MS}$) and the low-resolution version of the PAN image ($P^{L}$) as:(3)$$G^{i}=\frac{\tilde{MS}}{P^{L}}.$$

This pixel-wise injection scheme is called the Spectral Distortion Minimisation (SDM) model, which is performed especially for the GLP-SDM [23] and FBP-SDM [6]. Other examples of pixel-based pansharpening techniques are the Brovey transform (BT) [13] and Smoothing Filter-based Intensity Modulation (SFIM) [52].

#### 2.1.3. Projective Injection Scheme

This model consists of computing statistically the gain coefficients through a regression analysis [25] as:(4)$$G^{i}=\frac{cov({\tilde{MS}}^{i},P_{L}^{i})}{P_{L}^{i}},i=1,..,N_{B}.$$
where cov(.,.) and var(.) stand for the covariance and the variance, respectively.

In the context of pansharpening, this strategy was initially applied to CS methods within the GSA approach [17] in a band-wise manner. Then, it was adopted for MRA approaches, particularly for the GLP technique, in which it is processed at band-to-band [25] and at region level [11,24,45].

To summarize, the estimation of injection gains is a crucial step for pansharpening. It can be performed either globally (i.e., the same coefficient is applied to the whole image or to the different spectral bands), or locally (i.e., spatially variant coefficients are applied computed locally on regions).

### 2.2. Simple Linear Iterative Clustering Method

The main goal of superpixel segmentation is to subdivide an image into a large number of regions, such that each region lies within object boundaries defined over the image to be segmented. This process is known as image over-segmentation [46]. Among the popular superpixel segmentation methods, examples include Simple Linear Iterative Clustering (SLIC) [46], which belongs to the gradient-ascent based category. The algorithm starts from initial rough groups of pixels (i.e., clusters) obtained by means of the local k-means clustering method and iteratively refines them to provide better segmentation until satisfying certain convergence criteria. The approach can be applied for both greyscale and color images. In our study, a colored image is considered. A five-dimensional vector ${\left[l,a,b,x,y\right]}^{T}$ is defined to represent the features of a cluster, where the superscript ${\left[.\right]}^{T}$ denotes the transpose operator. ${\left[l,a,b\right]}^{T}$ is the color feature vector corresponding to the CIELAB color space and ${\left[x,y\right]}^{T}$ denotes the spatial feature vector. The main steps of SLIC are as follows:Initialize the *k* cluster centres defined by $C_{k}={\left[I_{k},x_{k},y_{k}\right]}^{T}$ which denotes the pixel position. The pixel intensity $I_{k}$ is incorporated into the feature vector $C_{k}$. The grid interval $S=\sqrt{\left(\frac{N}{k}\right)}$ approximately and uniformly determines the superpixels size, where *N* is the number of pixels.Each pixel at the position $\left(x_{i},y_{i}\right)$ is associated to the nearest cluster centre ($C_{k}$) at the position $\left(x_{k},y_{k}\right)$ according to the distance (Ds) estimated by the formula:(5)$$D_{s}=\sqrt{{\left(x_{i}-x_{k}\right)}^{2}+{\left(y_{i}-y_{k}\right)}^{2}}$$
and the color CIELAB distance $D_{c}$ is defined as:(6)$$D_{s}=\sqrt{{\left(l_{i}-l_{k}\right)}^{2}+{\left(a_{i}-a_{k}\right)}^{2}+{\left(b_{i}-b_{k}\right)}^{2}}$$
such that the clustering metric can be formulated:(7)$$D_{sc}=\sqrt{D_{c}^{2}+\frac{D_{s}^{2}}{S^{2}}m^{2}}$$
where *m* is the compactness parameter.By specifying the clustering metric $D_{sc}$ as in Equation (Equation 7), the parameter *m* allows computation of the relative interest between the grayscale similarity and spatial proximity.

In summary, the clustering steps provide a number of $N_{s}$ superpixels. The higher the number of generated superpixels $N_{s}$, the finer the over-segmentation. The SLIC superpixel segmentation procedure is summarized in Algorithm 1.
**Algorithm** **1:** Pseudocode of SLIC method applied to MS image.**Input**: MS: MS image (R,G,B bands), *k*: number of initial clusters, *m*: compactness parameter.**Output**: $G_{RGB}$: Label matrix result of the SLIC applied on RGB input image.Represent the RGB image in the CIELAB space color: (R,G,B)←(l,a,b).Initialize *k* cluster centres $C_{k}=\left[I_{k},x_{k},y_{k}\right]$ by sampling pixels at a uniform grid *S*: $S\leftarrow \sqrt{(}\frac{N}{k})$, where $C_{k}$ is the feature vector, $I_{k}$ is the pixel intensity and $\left(x_{k},y_{k}\right)$ are its coordinates.**for*** each cluster centres $C_{k}\left(x_{k},y_{k}\right)$ and given a pixel coordiantes $\left(x_{i},y_{i}\right)$***do**((
   Compute the spatial distance ($D_{s}$) using Equation (Equation 5) as:
$$D_{s}\leftarrow \sqrt{{\left(x_{i}-x_{k}\right)}^{2}+{\left(y_{i}-y_{k}\right)}^{2}}$$
   Compute the spectral (color) distance ($D_{c}$) using Equation (Equation 6):
$$D_{c}\leftarrow \sqrt{{\left(l_{i}-l_{k}\right)}^{2}+{\left(a_{i}-a_{k}\right)}^{2}+{\left(b_{i}-b_{k}\right)}^{2}}$$
   Compute the clustering distance ($D_{sc}$) using Equation (Equation 7):
$$D_{sc}\leftarrow \sqrt{D_{c}^{2}+\frac{D_{s}^{2}}{S^{2}}m^{2}}$$**end**(Compute the new cluster centre $C_{k}^{\prime }$ according to the lowest distance $D_{sc}$ estimated in the grid (2S×2S)Update the SLIC map by computing the distances between the previous $C_{k}$
$\left(D_{sc}\right)$ and the newest cluster $C_{k}^{\prime }$ ($D_{sc}^{\prime }$).Repeat the algorithm steps (3)-(5) until the distance $\epsilon_{sc}$ between two cluster centres is not a null-value: $\epsilon_{sc}\leftarrow D_{sc}-D_{sc}^{\prime }\neq 0$Label the resulting semantic segmentation map, $G_{RGB}$, assigning each superpixel with $l=\left\{1,..,L\right\}$.

### 2.3. Image Graph Representation via Region Adjacency Graph

Given an image I, it can be partitioned into distinct non-overlapping and non-empty regions (i.e., segments). For a positive integer n>1, we define ${\left\{R_{i}\right\}}_{i=1}^{n}$ as the finite set corresponding to the resulting segmented image, where the subscript ${\left(.\right)}_{i}$ indicates that the segment is labeled with an index i∈{1,…,n}, where *n* is the total number of segments. The segmentation process satisfies the following two relations:(8)$$I=\overset{n}{\underset{i=1}{⋃}}R_{i},$$
and(9)$$R_{i}\cap R_{j}=⌀\mathrm{for}\;i\neq j.$$

Let $G_{I}=\left(V,E\right)$ denote an indirect image-driven graph of an image I, which is called the Region Adjacency Graph (RAG) [48]. By definition, the RAG represents the adjacency of regions with a graph, which can be represented by a set of vertices (nodes) v∈V and a set of edges $e\in E\subseteq V^{2}$. Mathematically, the graph can be formalized as:(10)$$G_{I}=\left\{\left(v,e\right)\setminus v\in V\;and\;e\in E\right\}$$

In the context of image segmentation, V is the set of resulting regions (further the set of superpixels) from the image I and E is the set of all pairs of adjacent segments linked by a neighborhood relationship. Accordingly, they can be expressed as follows:(11)$$V={\left\{R_{i}\right\}}_{i=1}^{n},$$
and(12)$$E=\{e\setminus \forall e,\exists \left(v_{p},v_{q}\right)\in R^{2}/e=\left(v_{p},v_{q}\right)\}.$$

The RAG steps are represented in Algorithm 2. An example of an image-driven graph is represented graphically and displayed in Figure 1. It contains seven nodes (i.e., regions) and 10 edges. The RAG [cf. Figure 1b] is constructed from an image segmentation [cf. Figure 1a] such that black dots are the nodes ${\left\{v_{i}\right\}}_{i=1}^{7}$ and bloc lines are edges ${\left\{e_{j}\right\}}_{j=1}^{10}$.
**Algorithm ** **2:** Pseudocode of RAG method.**Input**: I: A greyscale image: $I\leftarrow G_{RGB}$.**Output**: $G_{I}=\left(V,E\right)$: RAG image-based graph: $G_{I}\leftarrow G_{RGB}$.Define a set of vertices V from an over-segmented image I using Equation (Equation 11).**for*** each couple of labels (p,q)***do**((
   Define the respective vertices as:
$$\left(vp,v_{q}\right)\leftarrow \left(R_{p},R_{q}\right)$$
   Compute the the edge between $\left(R_{p},R_{q}\right)$ using Equation (Equation 12).
**end**(Compute new vertices and edges $E_{pq}$ by repeating steps (1) et (2) until the distance between two edges is equal to zero, as: $E_{pq}\leftarrow \left(e_{p}-e_{q}\right)=0$Obtain the RAG segmented image defined by Equation (Equation 10).

## 3. Materials and Methods

The proposed graph-based pansharpening approach can be summarized in terms of the following three phases: image segmentation, gain estimation, and fusion, which are detailed in the subsections below. A schematic diagram of the proposed approach is presented in Figure 2.

**Figure 2 sensors-25-04992-f002:**
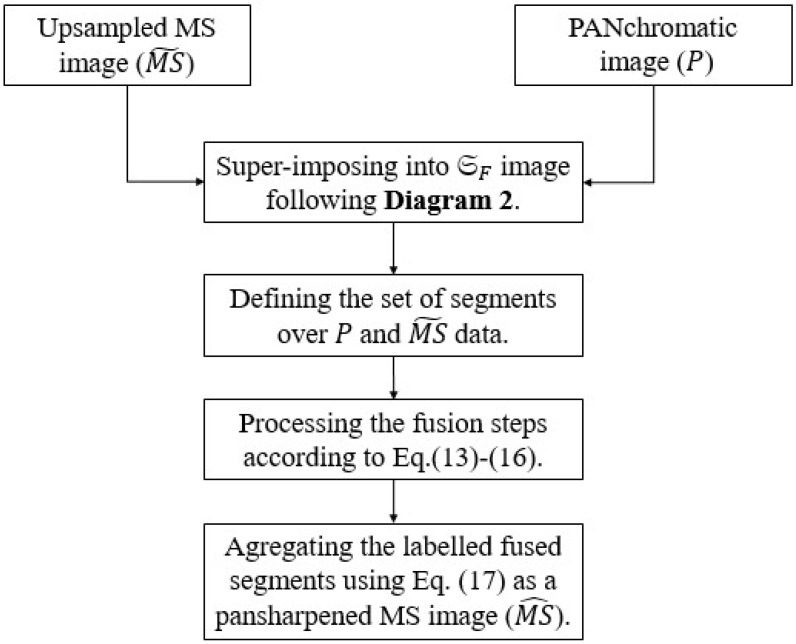
Overall diagram of the proposed graph-based pansharpening approach. Note that Diagram 2 is displayed in Figure 3.

**Figure 3 sensors-25-04992-f003:**
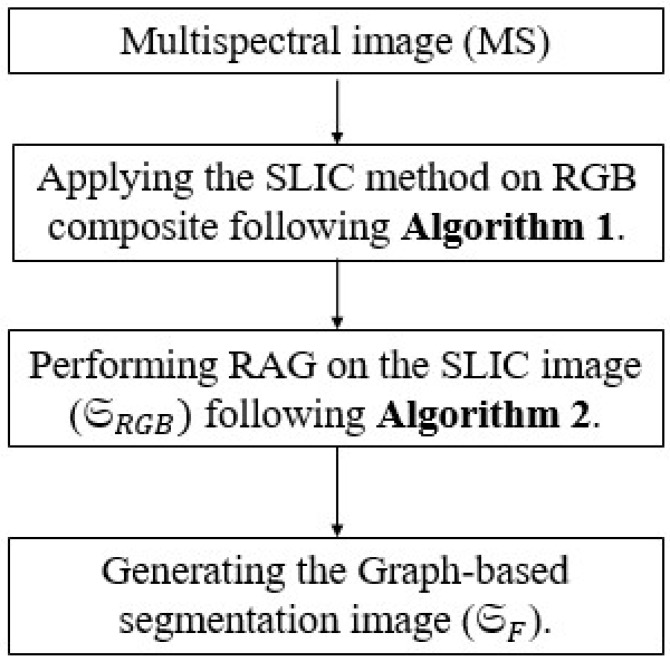
The steps involved in generation of the guidance image as a graph-based segmentation map.

### 3.1. RAG-Based SLIC Performed on MS Image

The main steps of the graph-based segmentation map are reported in Figure 3 and are experimentally illustrated in Figure 4. The image segmentation phase deals with the generation of graph-based superpixels using region similarities. The principle is to combine the superpixels, generated by means of SLIC, and the RAG concept to obtain novel graph-based superpixel entities. More specifically, it consists of merging each two adjacent regions in order to produce a novel region that contains all similar pixels from both of them. The process is iterated until fusing of similar region pairs occurs. The process is presented in more detail below.

#### 3.1.1. SLIC Segmentation

An over-segmentation algorithm is applied to the MS image by means of SLIC (only the RGB bands are considered) [46]. This allows transforming the input image, denoted as $S_{RGB}$, by aggregating homogeneous neighbouring pixels into a set of perceptually meaningful superpixels showing similar properties, which is the spectral information in our case study. A unique label is therefore assigned to each region.

#### 3.1.2. RAG on SLIC

The MS image is over-segmented into a set of superpixels, $S_{RGB}$. A RAG is built from the set of superpixels $S_{RGB}$ and denoted by $G_{RGB}=\left(V_{RGB},E_{RGB}\right)$, where $V_{RGB}={\left\{R_{i}\right\}}_{i}^{n}$ represents the set of resulting regions such that each region in the segmented image is a node in the graph and $E_{RGB}=\left(R_{i},R_{j}\right)$ is obtained as the joint edge linking two adjacent regions.

#### 3.1.3. Node Merging

This step enables improvement of the initial segmentation with the aid of the RAG. To this end, a hierarchical threshold cut method is performed. It enables removing the edges below a certain threshold and assigning a label to each connected component in the image-based graph. The threshold γ is determined as follows. Starting from an initial number of superpixels derived from the SLIC algorithm, merging between adjacent regions is performed iteratively in accordance with the greyscale pixels values (i.e., spectral information from the MS bands). γ is therefore defined as the value at which fusion between two adjacent vertices is completed. This procedure allows aggregating similar regions to obtain a uniform region.

In this study, the threshold is determined as follows. Given two vertices $v_{1}$ and $v_{2}$ linked by an edge *e* in the graph $G_{RGB}$, the merging weight $w_{e}$ represents the probability of merging these two nodes as a region. First, the centroid of each region is determined. Then, edges linking every pair of adjacent regions are highlighted. Next, edge weights are computed between every two nodes, which are defined as the difference in the average color between two regions. The more similar the regions are, the lower the weight between them.

Figure 5 depicts an example for determining the threshold parameter using the GeoEye-1 (Captown-GE1) and WorldView-3 (Adelaide-WV3) datasets. It shows the variation in the number of regions (ie., superpixels initially generated by SLIC) versus the grayscale pixel values of the considered images. It is observed that the number of regions decreases after the node merging stage by applying the RAG concept. Using a number of initial clusters $N_{c}=300$, the parameters for the resulting graph-based superpixels are equal to the following: $N_{s}=205$, $N_{g}=150$, and γ=50 for the Captown-GE1 dataset; and $N_{s}=200$, $N_{g}=120$, and γ=100 for the Adelaide-WV3 dataset. Moreover, we note that the second dataset gives a lower number of graph-based regions than the first dataset. This is explained by the fact that the urban dataset contains more textured and heterogeneous structures.

The proposed graph-based superpixel segmentation structure is shown graphically in Figure 4 using two samples of remote sensing datasets acquired from the GeoEye-1 [see Figure 4a–d] (GE1) and WorldView-3 (WV-3) [see Figure 4e–h] satellites. In this example, the corresponding boundaries, resulting from an initial segmentation method using SLIC are marked in blue color. The edges are marked as yellow lines and the centroids as red points. Comparing the obtained graph-based image with the segmentation results from directly applying SLIC [46], as in Figure 4d,h, we note that the presented process (RAG on SLIC over-segmentation) enables grouping of superpixels with similar sizes, as well as merging of small-sized superpixels. Recall that SLIC is considered an initial step for the proposed graph-based superpixel method. As depicted in Figure 5, the number of regions (i.e., graph-based superpixels) decreased from 220 to 160 for the Captown-GE1 dataset and from 200 to 105 for the Adelaide-WV3 dataset after the node merging step.

### 3.2. Graph-Based Superpixel Guided Regression Fusion Rule

Once the MS image is over-segmented into a set of graph-based superpixels, $G_{F}$, the latter is super-imposed into the up-sampled MS and the detail images, yielding sets of regions $R^{\tilde{MS}\left(j\right)}$ and $R^{D(j)}$ for each respective image. Furthermore, from the graph $G_{F}$, we computed the set of final regions (defined by $N_{g}$) to e involved in the local pansharpening process and consequently in the estimation of local injection factors. The pansharpening task consists in locally merging the PAN and MS regions in accordance with the GLP framework, and then computing the injection gains for each region defined by the graph-driven map $G_{F}$. More especially, this region-dependent fusion problem can be defined by the following model [7]:(13)$${\hat{ms}}_{i}^{\left(j\right)}=f\left(x_{i}^{\left(j\right)},{\hat{g}}_{i}^{\left(j\right)}\right),\;j=\left\{1,\ldots ,L\right\}.$$
where f(.,.) assigns the fusion process in accordance with the pyramidal framework, and the superscript ^*(j)*^ and *L* denote the label numbers. $x_{i}^{\left(j\right)}=\left[{\tilde{m}}_{i}^{\left(j\right)},d_{i}^{\left(j\right)}\right]$ represents the up-sampled multispectral and detail segments, respectively. ${\hat{g}}_{i}^{\left(j\right)}$ is the gain factor to be predicted.

The set of gain values are defined as a group of $N_{g}$ optimal solutions using the Simplex algorithm [7], which is given for each region of the label index $j=\{1,\ldots ,N_{g}\}$ as:(14)$${\hat{g}}_{i}^{\left(j\right)}=\underset{g_{i}^{\left(j\right)}}{\arg\max}\left(Q2^{n}\left({\hat{m}}_{i}^{\left(j\right)},{\bar{m}}_{i}^{\left(j\right)}\right)\right).$$
where $Q2^{n}$ is the Quaternion-based index [1,53] applied as an optimizer parameter used both for evaluation and fusion procedures, which are performed over all labeled segments in order to obtain the optimal gain values. argmax denotes the maximization solver. Notice that the $Q2^{n}$ formula (Equation (Equation 18)) is described in the “Quality Metrics” (Section 4.4). Equation (Equation 14) indicates that all pixels belonging to the same segment have the same injection gain ($\hat{g}$).

Next, for each spectral band, the set of gain values ${\left\{g_{i}^{\left(j\right)}\right\}}_{i=1}^{N_{g}}$ are spatially concatenated according to their label index to obtain meaningful gain maps:(15)$${\hat{G}}_{i}=cat\left({\hat{g}}_{i}^{\left(1\right)},...,{\hat{g}}_{i}^{\left(L\right)}\right).$$

The optimal injection coefficients are used to modulate the detail segments before inferences into the corresponding expanded MS segments.

The high-resolution MS segments are synthesized as the spatially arranged MS fused regions, ${\left\{{\hat{ms}}_{i}^{\left(j\right)}\right\}}_{i=1}^{N_{g}}$, by applying Equation (Equation 13) locally:(16)$${\hat{MS}}_{i}=cat\left({\hat{ms}}_{i}^{\left(1\right)},...,{\hat{ms}}_{i}^{\left(L\right)}\right),$$
and therefore Equation (Equation 16) can be written conventionally as follows:(17)$${\hat{MS}}^{i}={\tilde{MS}}^{i}+{\hat{G}}^{i}\circ D^{i},i=\left\{1,..,N\right\},$$

## 4. Experiments and Results

The experimental results, including methods and implementation details, datasets, and quality metrics are presented and discussed in this section.

### 4.1. Benchmarks and Implementation Details

Both pixel- and region-based pansharpening approaches are evaluated, and the effectiveness of the proposal is demonstrated. As a yardstick, interpolation of the original MS image using a 23-tap polynomial filter (EXP) [22] is utilized for the tests. Among the CS-based techniques, the following methods are considered: GSA [17], BDSD [19], PRACS [18], GSA-BPT [45], and BT-H [14]. With respect to MRA methods, AWLP [29], MF-HG [28], the GLP technique that adopts additive injection schemes (GLP-MTF) [22], multiplicative injection (GLP-SDM) [23], projective injection schemes (GLP-OLS) [25], cluster injection (GLP-BPT) [45], and our proposed model GraphGLP are used in comparative experiments. For VO-based techniques, CS-D [39], PWMBF [36], TV [38], and FE-HPM [40] are applied. Deep learning approaches include PNN [33,43] and CNN [40] for comparison purposes. For all the mentioned pansharpening methods, both reduced, in agreement with the Wald [44] and Zou protocols [54], and full resolution, in accordance with the QNR protocol [55], were tested and experimentally evaluated.

### 4.2. Implementation Details

The number of clusters $N_{c}$ is equal to 500 generated from the graph-based superpixel method. The compactness parameter (*m*) is set to $10^{-4}$. The value of the threshold γ is set to 400 for the two datasets, respectively. The resulting regions for the graph-based superpixels are equal to: $N_{s}=205$, $N_{g}=150$, and γ=50 for the Captown-GE1 dataset; and $N_{s}=200$, $N_{g}=120$, and γ=100 for the Adelaide-WV3 dataset.

### 4.3. Datasets

Two data sets were considered for the experiments in this study, in which two scenarios were acquired by two different sensors. The image sets were downloaded from the following website: “https://apollomapping.com/” (accessed on 2 April 2025). The data present two cases in terms of the number of spectral bands that characterize the original MS images (4 or 8 bands) and ensure diversity of the observed areas (e.g., vegetation and trees, roads, and buildings). The assessment of the two datasets is provided at reduced- and full-scale in agreement with the “quality metrics” described in Section 4.4. Details of these image sets are presented below.

1.Captown dataset: This dataset is displayed in Figure 6a,b. It represents a sub-urban area of the city of Capetown (South Africa). It is collected by the GeoEye-1 (GE-1) satellite that works in the visible and near-infrared (NIR) spectrum range. It is composed of a PAN and a four-band MS image (red, green, blue, and NIR). The spatial sampling interval (SSI) is 2 m for MS and 0.5 m for PAN images. The size of the dataset is 256×256×4 for an MS image and 1024×1024 for PAN. The resolution ratio is equal to four. The radiometric resolution is 16 bits. The characteristics of the Captown set include buildings, vegetation, and grass soil.2.Adelaide dataset: This dataset is exhibited in Figure 6c,d. It displays an urban area of Adelaide city (Australia). It is acquired by the WorldView-3 (WV-3) satellite, working in the visible and near-infrared (NIR) spectrum range. The data contain an eight-band MS image covering four standard (blue, green, red, and NIR1) and four new (coastal, yellow, red edge, and NIR2) bands and a PAN image, with a 1.2 m and a 0.3 m SSI, respectively. The size of the PAN image is 1024×1024 pixels and that of the MS image is 256×256×8 pixels. The resolution ratio is equal to four. The radiometric resolution is 16 bits. The characteristics of the Adelaide set include dense buildings, trees, soil, and some cars.

### 4.4. Quality Metrics

The proposed approach was evaluated by carrying out the reduced [44] and the full-resolution [55] assessment procedures. The Spectral Angle Mapper (SAM) [56], the Erreur Relative Globale Adimensionnelle de Synthèse (ERGAS) [44], the Spatial Correlation Coefficient (SCC) [54], the Structural Similarity Index Measure (SSIM) [57], the Peak Signal-to-Noise Ratio (PSNR) [58] and the Quaternion-based index ($Q2^{n}$) [55] are selected for the reduced scale assessment procedure and the Quality with no Reference (QNR) [55] for the full-scale quality assessment.

To quantify the quality of the fused products, we computed seven distinct measures:1.The Quaternion-based index, $Q2^{n}$, is a global multiband extension [1] of the Universal image quality index [53]. The quality metric was first introduced for evaluating a four-band pansharpened MS image, and extended to MS images with $2^{n}$ bands. Taking an image with *N* spectral bands, a pixel is defined by a hypercomplex number having one real part and N−1 imaginary parts. Let us consider two hypercomplex numbers z=z(u,v) and $\hat{z}=\hat{z}\left(u,v\right)$, which represent the reference and the fused spectral images at pixel coordinates (u,v). The $Q2^{n}$ index is written as follows:(18)$$Q2^{n}=\frac{\left|\sigma_{z\hat{z}}\right|}{\sigma_{z}\sigma_{\hat{z}}}\frac{2\sigma_{z}\sigma_{\hat{z}}}{\sigma_{z}^{2}+\sigma_{\hat{z}}^{2}}\frac{2\left|\bar{z}\right|\left|\bar{\hat{z}}\right|}{{\left|\bar{z}\right|}^{2}{\left|\bar{\hat{z}}\right|}^{2}}$$
where $\sigma_{z\hat{z}}$ denotes the covariance between the two hypercomplex number *z* and $\hat{z}$. $\sigma_{z}$ (similarly $\sigma_{\hat{z}}$) denotes the standard deviation of the hypercomplex number *z* (similarly $\hat{z}$). |z| (similarly $\left|\hat{z}\right|$) denotes the mean value of the hypercomplex number *z* (similarly $\hat{z}$). The $Q2^{n}$ metric is viewed as the product of three terms (as their ranking), which are the hypercomplex correlation coefficient, the contrast changes, and the mean bias between *z* and $\hat{z}$.2.The Spectral Angle Mapper (SAM) is a global spectral dissimilarity or distortion index. It was proposed for discriminating the materials based on their reflectance spectra [56]. Given two vectors, $v=\left[v_{1},\ldots ,v_{N}\right]$ and $\hat{v}=\left[{\hat{v}}_{1},\ldots ,{\hat{v}}_{N}\right]$, which define the reference spectral pixel vector and the corresponding fused spectral pixel, respectively. The SAM index denotes the absolute value of the spectral angles between the two vectors, given by:(19)$$SAM=\arccos(\frac{\langle v,\hat{v}\rangle }{{∥v∥}_{2}.{∥\hat{v}∥}_{2}})$$The SAM metric is expressed in degrees (^∘^), and its optimal value is equal to zero if the reference and the fused vectors are spectrally identical.3.The Erreur Relative Globale Adimensionnelle de Synthèse (ERGAS) [44] is a normalized dissimilarity index estimated between the reference and the multi-band fused images. It is defined as:(20)$$ERGAS=100\frac{d_{h}}{d_{l}}\sqrt{\frac{1}{N}}\sum_{n}^{N}\frac{RMSE{\left(n\right)}^{2}}{\mu (n)}$$
where $\frac{d_{h}}{d_{l}}$ is the ratio between the pixel size of PAN and MS (generally equal to $\frac{1}{4}$ for many sensors in the pansharpening case study). μ(n) is the mean (i.e., average) of the $n^{th}$ band of the reference image. *N* is the number of bands. RMSE is the root mean square error of the $n^{th}$ band. Low values of ERGAS indicate high similarity between the reference and the fused images.4.The Spatial Correlation Coefficient (SCC) is used for assessing the similarity between the spatial details of the pansharpened and the reference images. It uses a high-pass filter (a Laplacian filter) to extract the high-frequency information from both the reference and fused images, and computes the correlation coefficient between them. It is usually known as the Zou protocol [54]. The kernel of the Laplacian filter involved in this procedure is given by:(21)$$F=\left(\begin{array}{ccc} -1 & -1 & -1\\ -1 & 8 & -1\\ -1 & -1 & -1 \end{array}\right)$$
and the Pearson correlation coefficient is computed as:(22)$$CC=\frac{\sum_{i=1}^{w}\sum_{j=1}^{h}\left(X_{ij}-\mu_{X}\right)\left(Y_{ij}-\mu_{Y}\right)}{\sum_{i=1}^{w}\sum_{j=1}^{h}{\left(X_{ij}-\mu_{X}\right)}^{2}{\left(Y_{ij}-\mu_{Y}\right)}^{2}}$$
where *X* and *Y* are the reference and fused images in pansharpening, respectively. *w* and *h* are the width and height of the image. μ represents the average of the image *X* (similarly the image *Y*). A high SCC value indicates that most of the spatial details of the PAN image are injected into MS bands during the fusion process.5.The Structural Similarity Index Measure (SSIM) [57] estimates the overall fusion quality by computing the first-order statistics (the mean, variance, and covariance) of the reference and its corresponding fused MS image. The SSIM index is the combination of three contrast modules, called brightness, contrast, and structure, given by:(23)$$SSIM\left(X,Y\right)=\left[l{\left(X,Y\right)}^{\alpha }c{\left(X,Y\right)}^{\beta }s{\left(X,Y\right)}^{\gamma }\right]$$
where(24)$$\left\{\begin{array}{c} l\left(X,Y\right)=\frac{2\mu_{x}\mu_{y}+c_{1}}{\mu_{x}^{2}+\mu_{y}^{2}+c_{1}}\\ c\left(X,Y\right)=\frac{2\delta_{x}\delta_{y}+c_{2}}{\delta_{x}^{2}+\delta_{y}^{2}+c_{2}}\\ s\left(X,Y\right)=\frac{2\delta_{xy}+c_{3}}{\mu_{x}\mu_{y}+c_{3}} \end{array}\right.$$
and $\mu_{x}$, $\mu_{y}$, $\delta_{x}^{2}$, $\delta_{y}^{2}$, and $\delta_{x}y$ are the means, variances, and covariances of *X* and *Y*, respectively.The optimal value of SSIM is equal to one; a high value indicates high similarity between the images.6.The Peak Signal-to-Noise Ratio (PSNR) is originally introduced to estimate the noise quantity in images [57]. Then, it is used to calculate the Mean Squared Error (MSE) between the original and distorted images. It is expressed as:(25)$$PSNR=10\log_{10}\frac{MAX_{I}^{2}}{MSE}$$
where $MAX_{I}$ denotes the maximum value representing the color of pixel of an image *I*.In the context of image pansharpening, MSE is defined as:(26)$$MSE=\frac{1}{mn}\sum_{i=0}^{m-1}\sum_{j=0}^{n-1}{∥MS\left(i,j\right)-\hat{MS}∥}^{2}$$
where MS and $\hat{MS}$ are the reference and pansharpened images of size m×n.The high value of PSNR indicates that the reconstructed image presents less distortion compared to the reference MS image.7.The Quality with no Reference (QNR) protocol [55] is proposed for the assessment of fused images at full-resolution at the scale of the PAN band. This metric allows tackling the problem of unavailability of the true MS image for a full-scale PAN image.The QNR index is computed as the product of two normalized measurements of the spectral and spatial consistencies. In more detail, two distortions indexes, called the spectral distortion $D_{\lambda }$ and the spatial distortion $D_{s}$, are combined to define the QNR index, as follows:(27)$$QNR={\left(1-D_{\lambda }\right)}^{\alpha }{\left(1-D_{s}\right)}^{\beta }$$
where α=β=1 (for most applications).The optimal value of QNR is equal to one.

### 4.5. Results and Discussion

#### 4.5.1. General Fusion Results

The fusion results of the pansharpening methods on the Captown and Adelaide data at reduced resolution are displayed in Figure 7 and Figure 8. An enlarged area is displayed at full-scale evaluation (cf. Figure 9 and Figure 10). The comparative pansharpening pipelines are composed of several categories of methods: traditional, such as MRA and CS techniques (e.g., BDSD, PRACS, GSA, AWLP, MFHG, and GLP), second-generation methods based on physical models of sensors (e.g., FE-HPM, PWMBF, GLP-MTF, and TV), and de-hazing methods using radiative transfer through the atmosphere (e.g., BT-H).

The experiments evaluate the advantages of the adaptive multi-resolution pansharpening approach with injection rules using the ordinary least squares (known as regression) model and the optimized simplex method. Accordingly, the method is called GraphGLP-OLS for the first model, and GraphGLP-Simplex for the second case. The results were analysed in accordance with the statistical criteria and the spatial-spectral characteristics of the image sets to take advantage of the variability in the datasets in terms of pixel reflectances and textures [24,59]. GraphGLP-Simplex outperforms its statistical variants, GraphGLP-OLS and GLP-OLS, for most quality indexes because it takes into consideration the pixel neighbors of the segment in task. The variational-optimization (PMBF, TV, FE-HPM) and the deep learning (PNN and CNN) methods are included in quality assessment (cf. Figure 7, Figure 8, Figure 9 and Figure 10). It is clearly seen that the PNN results present a pixelization or blurring effect for both the GeoEye1 and WorldView-3 datasets. The CNN results display a high spectral quality compared to the up-sampled MS image, which could be explained by over-enhancement depending on the field of application. Analysis of the fused results confirms the diversity of the pansharpening methods which are crucial for validating the neW models. A compromise between visual analysis and quantitative assessment is established in this study, which may be explained by the fact that pansharpening can produce mathematically optimized results with respect to the computation complexity (such as in deep learning or sparse representation methods).

#### 4.5.2. Full-Scale Image Results

The full-resolution experiments for the GeoEye-1 and WorldView-3 datasets are illustrated in Figure 9 and Figure 10, respectively. For the GeoEye-1 dataset, the results provided by GSA, PRACS, and BDSD, depicted in Figure 9b–d, suffer from spectral distortions. An unnatural color is exhibited in the red building. The MRA-based methods [AWLP and MF-HG, displayed in Figure 9f,g, respectively] provide close spatial quality. Considering the methods in which the fusion is performed at region level, such as GLP-BPT, GSA-BPT, GraphGLP-OLS, and GraphGLP-simplex, it is noted that these context-adaptive methods better preserve the spectral as well as the spatial quality when compared to the pixel-based methods. Numerical results are shown in Table 1. Considering the Adelaide dataset, the fused products of GSA, BDSD, and PRACS [Figure 10d,e] show approximately similar contrast. Moreover, it is clearly observed that the AWLP and MF-HG [Figure 10f,g] produce high-resolution MS images with satisfactory spectral preservation. Compared to the adaptive approaches, such as GLP-BPT and GLP-SDM, the proposed GraphGLP-OLS and GraphGLP-simplex provide the best fusion results. The obtained results are confirmed by the quantitative experiments reported in Table 2.

**Figure 9 sensors-25-04992-f009:**
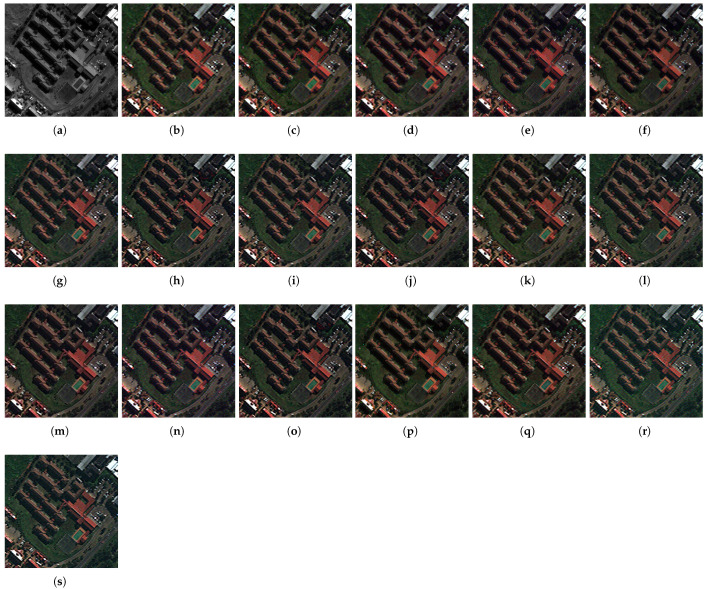
Enlarged close-ups of fused results at full-scale on the GeoEye-1 dataset. (**a**) PAN (**b**) MSup (**c**) BDSD (**d**) PRACS (**e**) GSA (**f**) GSA-BPT (**g**) AWLP (**h**) MFHG (**i**) GLP-MTF (**j**) GLP-SDM (**k**) GLP-BPT (**l**) GLP-OLS (**m**) PWMBF (**n**) TV (**o**) FE-HPM (**p**) BT-H (**q**) PNN (**r**) CNN (**s**) GraphGLP-simplex.

#### 4.5.3. Quantitative Performances

Table 1 and Table 2 summarize the fused results of the comparative evaluation of the proposed approach and the benchmarked pansharpening methods according to Wald’s protocol [e.g., ERGAS, SAM, SSIM, $Q2^{n}$ and PSNR], the Zou protocol (SCC), and the QNR protocol, for the two datasets. The proposed approach obtains satisfactory values in quality assessment. Furthermore, it is clearly shown that the performance of the fused results obtained with the regression scheme applied directly to the superpixels is lower than that obtained with the simplex-guided graph-based superpixel entities. Considering the level of detail injection, the fused products obtained at the region levels produce better results than those using an injection scheme at pixel level [23]. The proposed model gives enhanced fused results since it takes into account the spatial structural information defined by the graph structure of the data. The main limitation of the proposed approach is estimation of the injection weights at reduced resolution, in which the scale property is adopted according to Wald’s protocol [44]. Moreover, the injection coefficients are linearly estimated on segments, such that non-linear computation of gains could be advantageous for tracking cars or tackling the non-overlapping of MS bands [60].

**Figure 10 sensors-25-04992-f010:**
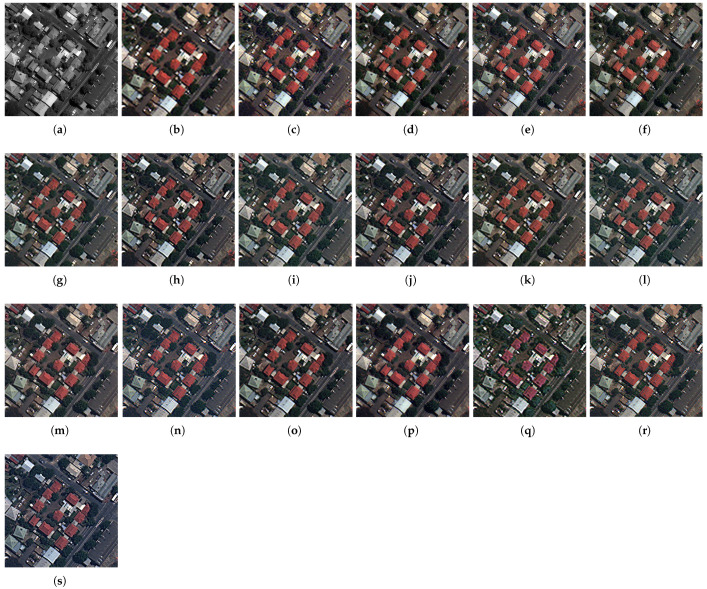
Enlarged close-ups of fused results at full-scale on the World-view-3 dataset. (**a**) PAN (**b**) MS up-sampled (**c**) BDSD (**d**) PRACS (**e**) GSA (**f**) GSA-BPT (**g**) AWLP (**h**) MFHG (**i**) GLP-MTF (**j**) GLP-SDM (**k**) GLP-BPT (**l**) GLP-OLS (**m**) PWMBF (**n**) TV (**o**) FE-HPM (**p**) BT-H (**q**) PNN (**r**) CNN (**s**) GraphGLP-simplex.

## 5. Conclusions

A novel region-based pansharpening method is proposed. The multispectral image is represented by a Region Adjacency Graph (RAG) structure. It is obtained from an initial over-segmentation by means of the SLIC method. The gains estimation and fusion phases are both performed over ensemble connected graph-based superpixels. The application of our proposed method involves first performing computation of the spectrally variant injection coefficients by means of a Simplex optimization model; then, introduction of a statistical ordinary least squares model during the fusion task.The fusion results are evaluated in comparison with traditional, variational optimization and deep learning-based pansharpening methods. The advantages of the proposed technique are analyzed and validated using two very-high-resolution remote sensing images collected by the GeoEye-1 and WorldView-3 satellites. Forthcoming research will consider application of the combination of gradient and graph-based superpixel segmentation to the pansharpening. Furthermore, estimation of object localization in the scene (e.g., cars, trees, or buildings) will require pixels spectra analysis.

## Figures and Tables

**Figure 1 sensors-25-04992-f001:**
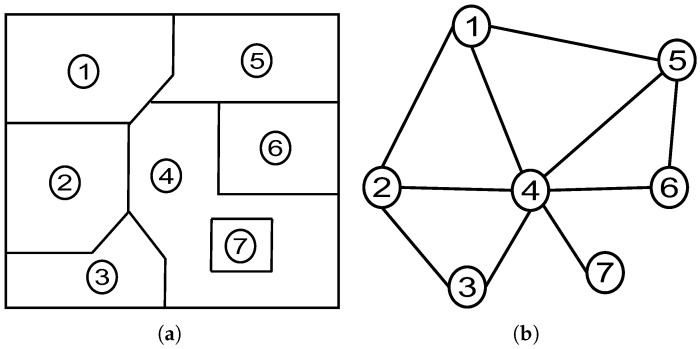
(**a**) Image partitioning and (**b**) Region Adjacency Graph (RAG). As example, the image is represented by seven numbered regions and delimited by ten edges (borders). The corresponding graph-based structure, RAG, is represented by the nodes relationships estimated between neighbour regions or segments.

**Figure 4 sensors-25-04992-f004:**
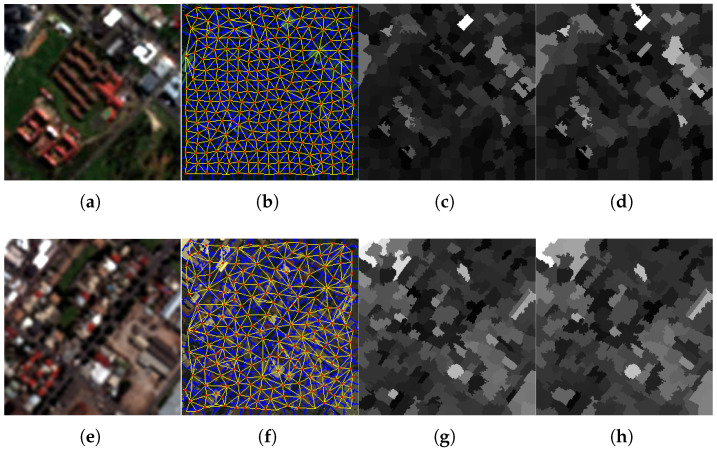
Illustration of generation of the guidance injection map. The first and second rows display the described process on GeoEye-1 and Adelaide datasets, respectively. The first column displays the expanded MS image. The second and third column show the initial RAG based on SLIC of the MS image and the segmented image, respectively. The final regions are overlaid on the source image.

**Figure 5 sensors-25-04992-f005:**
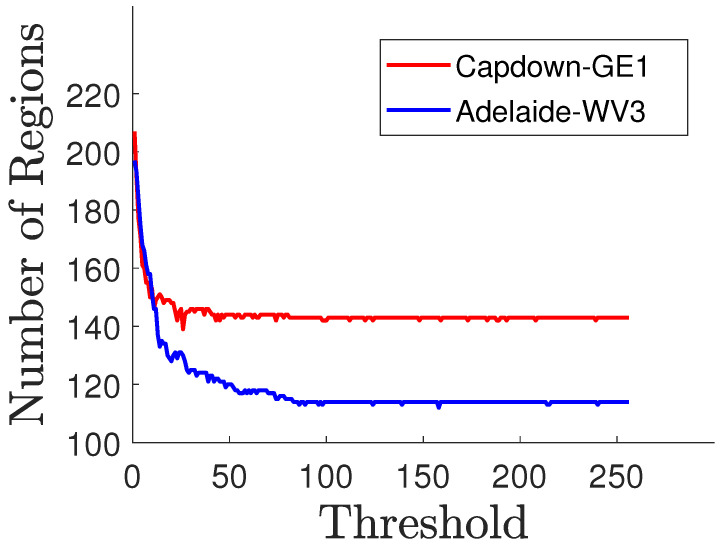
Threshold computing based on the RAG concept for $N_{c}=300$.

**Figure 6 sensors-25-04992-f006:**
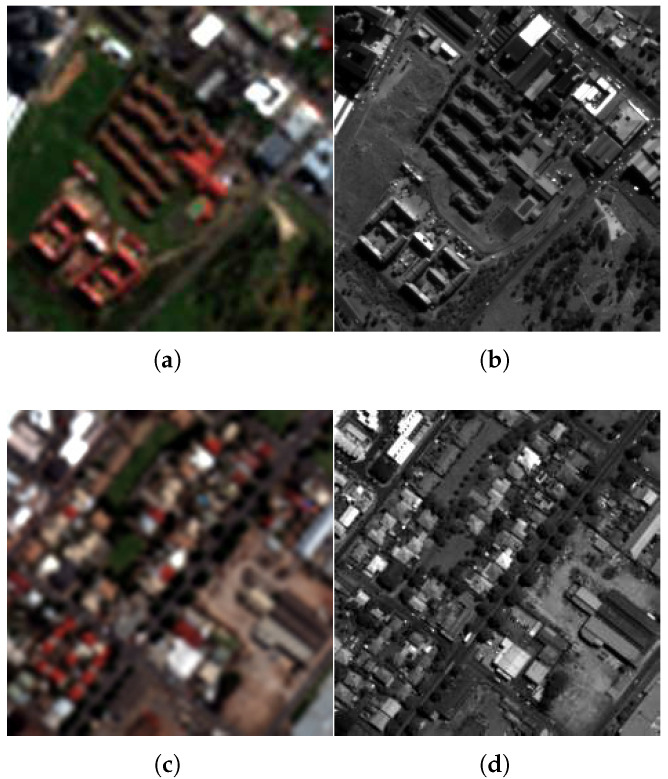
Datasets. (**a**,**b**) Captown dataset, (**c**,**d**) Adelaide dataset. The first column presents the expanded MS image, and the second column displays the PAN image.

**Figure 7 sensors-25-04992-f007:**
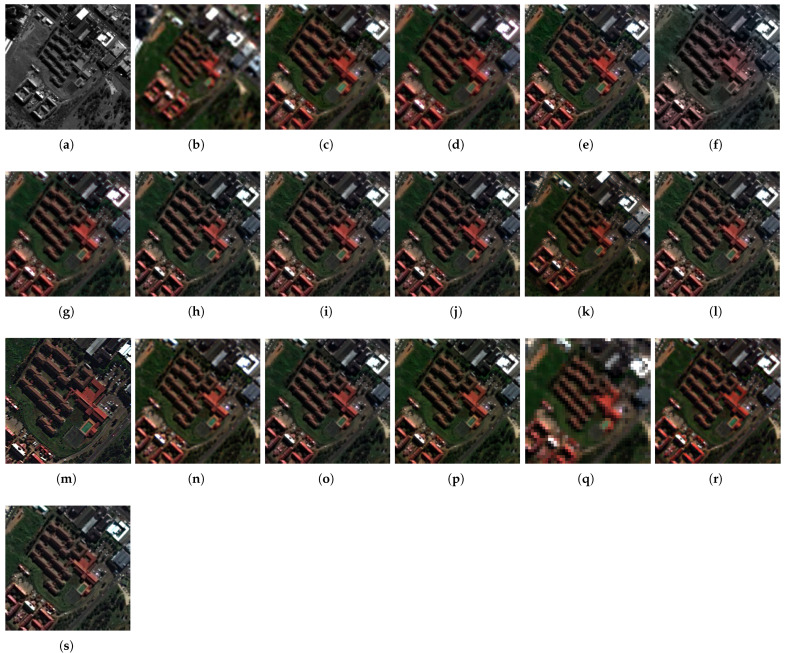
Fused results at reduced-scale on GeoEye-1 dataset. (**a**) PAN (**b**) MSup (**c**) BDSD (**d**) PRACS (**e**) GSA (**f**) GSA-BPT (**g**) AWLP (**h**) MFHG (**i**) GLP-MTF (**j**) GLP-SDM (**k**) GLP-BPT (**l**) GLP-OLS (**m**) PWMBF (**n**) TV (**o**) FE-HPM (**p**) BT-H (**q**) PNN (**r**) CNN (**s**) GraphGLP-simplex.

**Figure 8 sensors-25-04992-f008:**
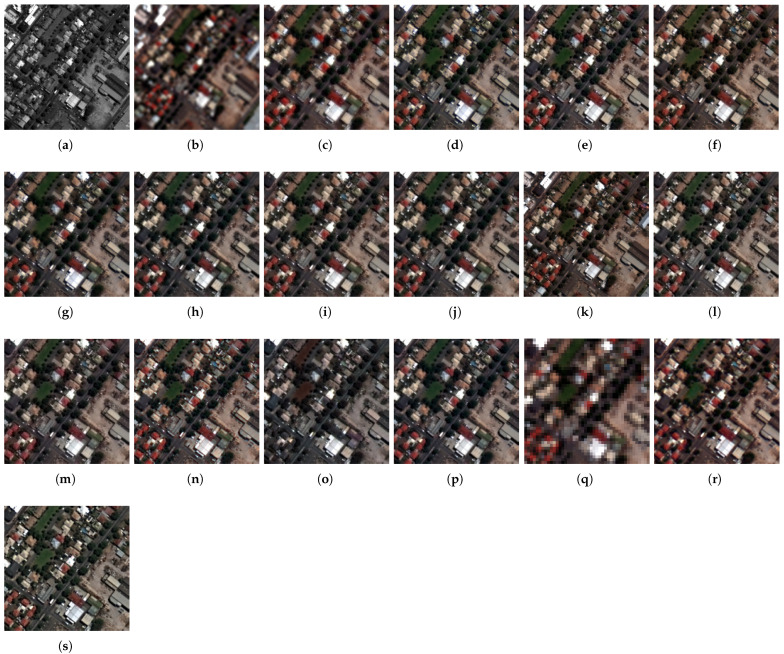
Fused results at reduced-scale on Worldview-3 dataset. (**a**) PAN (**b**) MS up-sampled (**c**) BDSD (**d**) PRACS (**e**) GSA (**f**) GSA-BPT (**g**) AWLP (**h**) MFHG (**i**) GLP-MTF (**j**) GLP-SDM (**k**) GLP-BPT (**l**) GLP-OLS (**m**) PWMBF (**n**) TV (**o**) FE-HPM (**p**) BT-H (**q**) PNN (**r**) CNN (**s**) GraphGLP-simplex.

**Table 1 sensors-25-04992-t001:** Performance indices of pansharpening algorithms for Captown images. **Bold:** the best.

Method	Q4	SAM	ERGAS	SCC	SSIM	PSNR	QNR
EXP	0.741	9.474	10.389	0.243	0.734	−39.318	0.921
BDSD	0.879	8.788	7.526	0.636	0.850	-36.985	0.926
PRACS	0.839	9.380	8.549	0.586	0.819	−37.751	0.931
GSA	0.853	9.456	7.888	0.637	0.837	−37.441	0.881
GSA-BPT	0.901	8.498	6.887	0.657	0.884	−36.047	0.897
AWLP	0.898	8.377	7.399	0.661	0.892	−36.085	0.919
MF-HG	0.911	8.762	6.656	0.676	0.874	−36.222	0.846
CS-D	0.861	12.557	7.961	0.661	0.810	−38.375	0.675
GLP-MTF	0.901	8.689	6.863	0.658	0.880	−36.261	0.839
GLP-SDM	0.912	8.812	6.465	0.682	0.875	−36.095	0.858
GLP-OLS	0.905	9.184	6.725	0.652	0.869	−36.504	0.835
GLP-BPT	0.902	8.492	6.823	0.656	0.886	−35.974	0.935
PWMBF	0.874	10.133	7.489	0.645	0.836	−37.550	0.764
TV	0.862	9.371	7.874	0.413	0.809	−37.910	**0.942**
FE-HPM	0.914	8.804	6.419	0.681	0.876	−36.066	0.839
BT-H	0.910	8.168	6.591	**0.690**	0.887	**−35.877**	0.913
PNN	0.889	**8.016**	6.905	0.624	0.873	−36.013	0.919
CNN	0.899	8.285	6.684	0.644	0.872	−35.992	0.941
GraphGLP-OLS	0.903	8.999	6.749	0.655	0.873	−36.392	0.935
GraphGLP-Simplex	**0.924**	8.954	**6.381**	0.665	**0.895**	−36.061	0.935

**Table 2 sensors-25-04992-t002:** Performance indices of pansharpening algorithms for Adelaide images. **Bold**: the best.

Method	Q8	SAM	ERGAS	SCC	SSIM	PSNR	QNR
EXP	0.706	8.205	8.121	0.249	0.601	−43.713	0.915
BDSD	0.845	7.957	6.161	0.646	0.770	−41.506	**0.943**
PRACS	0.847	8.174	5.871	0.665	0.794	−41.167	0.917
GSA	0.838	8.178	5.970	0.685	0.792	−41.326	0.884
GSA-BPT	0.885	7.811	5.266	0.709	0.834	−40.210	0.847
AWLP	0.875	7.950	5.466	0.704	0.820	−40.410	0.890
MF-HG	0.880	8.040	5.512	0.698	0.817	−40.661	0.835
CS-D	0.803	11.520	6.628	0.700	0.734	−42.486	0.660
GLP-MTF	0.880	7.956	5.391	0.703	0.822	−40.457	0.838
GLP-SDM	0.881	8.028	5.355	0.705	0.820	−40.441	0.845
GLP-OLS	**0.887**	8.174	5.250	0.698	0.830	−40.364	0.850
GLP-BPT	0.886	**7.795**	**5.236**	0.708	0.835	−40.164	0.879
PWMBF	0.855	8.985	5.788	0.701	0.796	−41.239	0.753
TV	0.895	7.853	5.206	0.708	**0.839**	−40.203	0.949
FE-HPM	0.882	8.030	5.347	0.705	0.821	−40.429	0.839
BT-H	0.894	8.049	5.225	0.713	0.835	−40.197	0.816
PNN	0.860	8.611	7.037	0.540	0.785	−42.137	0.908
CNN	0.763	8.910	7.283	0.403	0.651	−43.048	0.920
GraphGLP-OLS	0.886	8.187	5.260	0.697	0.829	−40.381	0.922
GraphGLP-Simplex	**0.909**	8.183	**4.935**	**0.861**	0.720	**−39.792**	0.922

## Data Availability

The data presented in this study are available on request from the corresponding author. All geospatial images involved in the experiments have been authorized by Mr. B. Adam McCarty.

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
