# Peer review of "A Graph-Based Superpixel Segmentation Approach Applied to Pansharpening"

_sensors, 2025, doi:10.3390/s25164992_

Round 1

Reviewer 1 Report

Comments and Suggestions for Authors

1.        The Preliminaries and Background section should include more related works on the technology, rather than focusing solely on technical descriptions. This would give the section a more academic tone.

2.        The methodology section, it is recommended to add a network diagram to aid in understanding the paper and its innovations, instead of relying solely on textual descriptions.

3.        The motivation behind the work, the highlights of the proposed method compared to existing methods, and the significance of the approach are not clearly stated in the abstract and introduction. It is important to clarify the problems that the method addresses.

4.        The comparison methods used in the experimental evaluation appear to be outdated. Updating the comparison methods to reflect more recent approaches in the field would be beneficial.

5.        The Captown dataset and Adelaide dataset used in this paper lack references.

6.        The Visual Analysis section needs to show more experimental results. In addition, it is suggested to highlight the key parts with red boxes to reflect the effectiveness of the proposed method.

7.        The calculation formulas for five quantitative indicators in the Quantitative Performances section need further explanation.

8.        Similarly, the discussion of related work seems insufficient. A more comprehensive review of current research in the field is recommended

9.        What are the disadvantages of the proposed method?

Comments on the Quality of English Language

No comments.

Reviewer 2 Report

Comments and Suggestions for Authors

1. In section 3.1.3, some key parameters were mentioned, such as Ns, Ng,γ. Suggest the author to provide a detailed explanation of the selection criteria for these parameters and their impact on the experimental results. Can experiments be added to demonstrate the changes infusion results under different parameter values to demonstrate the rationality of parameter selection.

2. In order to present the steps of each algorithm more clearly in Section 3 of the paper, it is recommended to add a detailed algorithm flowchart.

3. In section 4.1.1, the experimental result images should include more types of scenes. Currently, the experimental result display images are too single, and for some areas with rich details, they can be enlarged for a more intuitive comparison of the fusion effects of different methods.

4. In Section 4.3, in addition to existing evaluation metrics such as SAM, ERGAS, sCC, Q2n, and QNR, it is recommended to add some other commonly used evaluation metrics such as Peak Signal to Noise Ratio (PSNR) and Structural Similarity Index (SSIM) to more comprehensively evaluate the quality of the fusion results, and each evaluation metric should be briefly explained to explain its specific role in evaluating the fusion results.

5. In the comparison methods, it is recommended to add some of the latest methods, as the current comparison methods are not from the past 5 years. In addition, other methods can be compared in more detail with the relevant parts of the proposed method, reflecting the innovation and advantages of the proposed method.

Reviewer 3 Report

Comments and Suggestions for Authors

This paper presents a novel graph-based superpixel segmentation approach for pansharpening, which aims to optimally combine spatial information from high-resolution panchromatic (PAN) images and spectral information from low-resolution multispectral (MS) images to generate high-resolution MS images. The method involves using Simple Linear Iterative Clustering (SLIC) to over-segment the MS image, followed by a Region Adjacency Graph (RAG) merging stage to create a graph-based segmentation map. This map guides the injection of spatial details during the fusion process, with gain coefficients estimated using a regression analysis based on the Simplex algorithm. The proposed method was evaluated using GeoEye-1 and WorldView-3 datasets, demonstrating significant improvements in both spectral fidelity and spatial quality compared to traditional pixel-based and other region-based pansharpening techniques. The results highlight the effectiveness of the graph-driven approach in preserving local contextual information and enhancing the overall quality of the fused images.

1. The references are outdated. The latest reference is from 2021. The authors should add more recent literature from 2022 to 2024.

2. Following the above comment, more recent baselines should be added into Section 4. Experiments and Results.

3. The authors should at least give a short introduction to the baselines used in Section 4.1. Benchmarks and Implementation Details.

4. The evaluation metrics should be further explained in 4.3. Quality Metrics, better with mathematical equations.

5. The introduction provides a comprehensive overview of pansharpening techniques, but it could benefit from a clearer structure. Consider breaking down the introduction into subsections such as "Background," "Related Work," and "Motivation" to enhance readability.

6. The methodology section is detailed but can be improved by adding more visual aids (e.g., flowcharts or diagrams) to illustrate the steps involved in the proposed approach.

7. The description of the Simple Linear Iterative Clustering (SLIC) and Region Adjacency Graph (RAG) methods is thorough, but it might be helpful to include pseudocode for the key algorithms. 

8. Some of the mathematical equations (e.g., gain estimation and fusion rules) could be explained more clearly. Adding intermediate steps or examples would help readers follow the logic more easily.

9. The datasets used for experiments are described briefly, but more information about the specific characteristics of the images (e.g., types of land cover, specific features in the scenes) would be useful. 

10. The tables summarizing the performance metrics are comprehensive, but it would be useful to include a brief discussion of the significance of the results. For example, explaining why certain metrics are more important than others or how the proposed method outperforms others in specific aspects.

11. The paper mentions future perspectives briefly. It would be helpful to expand on potential applications or extensions of the proposed method, such as combining gradient-based and graph-based segmentation or exploring the use of machine learning techniques.

12. Finally, the authors should make sure that all abbreviations have been well defined and used afterwards. For example, RAG for Region Adjacency Graph should be used in the conclusion.

Round 2

Reviewer 1 Report

Comments and Suggestions for Authors

The author has solved my comments.

Reviewer 3 Report

Comments and Suggestions for Authors

No further comments.